# Improving Railway Alignment Selection in Mountainous Areas with Complex Vegetation: A Multisource Data Landslide Identification Approach for Assisted Decision-Making Research

**Jin Qian** [1,2], **Lei Li** [2,3], **Sitong Wu** [1], **Jinting Liu** [2,3] **and Yu Zhang** [2,3,*]

1 Institute of Computing Technologies, China Academy of Railway Sciences Corporation Limited, Beijing 100081, China; qianjin@rails.cn (J.Q.); w700710828@rails.cn (S.W.)
2 Key Laboratory of Railway Industry of Plateau Railway Transportation Intelligent Management and Control, Lanzhou Jiaotong University, Lanzhou 730070, China; 11210013@stu.lzjt.edu.cn (L.L.); liujinting0712@163.com (J.L.)
3 School of Traffic and Transportation, Lanzhou Jiaotong University, Lanzhou 730070, China
* Correspondence: zhangyu@mail.lzjtu.cn

**Abstract:** In order to provide important assistance for the scientific and effective route selection of future planned railways in the research area and to quickly and accurately identify the distribution range of landslides, thereby proactively mitigating the impact of geological hazards on railways under earthquake conditions, this study aims to shift the risk threshold for geological hazards and provide a scientific basis for the accurate planning and route selection of railways in mountainous areas. Jiuzhaigou was selected as the research area and postearthquake surface deformation information in the study area was obtained through Sentinel-1 satellite radar data. Based on Sentinel-2 optical remote sensing imagery, the changes in vegetation indices in the study area before and after the earthquake were analyzed in depth. The concept of vegetation index difference was proposed as a characteristic parameter for landslide information interpretation and a method combining surface deformation information was developed for landslide information interpretation. According to this method, the study area experienced a deformation subsidence of up to 14.93 cm under the influence of the earthquake, with some areas experiencing an uplift of approximately 6.0 cm. The vegetation index difference in the research area ranged from $-1.83502$ to $1.45366$. The total number of landslides extracted is 12.034 km$^2$ and 164 landslide points are marked, with an overall recognition accuracy of 92.6% and a Kappa coefficient of 0.876. The research results provide new research ideas for landslide information interpretation and can be used to assist in the decision-making of mountain railroad alignment options.

**Keywords:** landslide information; vegetation index; earthquake zone; remote sensing

## 1. Introduction

With the rapid development of China's railway industry, the construction of a developed and dense road network is urgent. However, the road network construction in western regions is still relatively weak. In the coming years, the investment and construction of railways in the central and western regions will inevitably increase. The rapid development of railways requires continuous improvement of survey and design levels, and railway route selection work should be more efficient and scientifically reasonable. Landslides can cause huge harm to railway construction and must be prevented from occurring to reduce the disasters they bring.

Landslide geological hazards are sudden, especially those triggered by earthquakes and mainly concentrated in mountainous areas. On 8 August 2017, a 7.0 magnitude earthquake occurred in Jiuzhaigou County, Aba Prefecture, Sichuan Province, with a depth of 20 km, triggering a large number of earthquake landslides accompanied by

many secondary hazards. According to incomplete statistics from the China Earthquake Administration [1], the earthquake caused serious casualties and property damage, with 25 people killed, over 170,000 people affected and over 70,000 houses damaged to varying degrees by August 13. Therefore, it is necessary to investigate the spatial distribution characteristics of landslides and interpret and analyze the landslide information after the earthquake.

At present, the two main types of conventional landslide survey methods are manual field surveys and interpretation based on optical remote sensing images [2]. Earthquake-induced landslides are widespread and numerous and the mountainous terrain is complex and extensive, making it time-consuming and labor-intensive to rely solely on conventional manual field surveys. Remote sensing technology has the distinct advantages of providing a macroscopic view, rapid data acquisition and wide coverage. This allows for quick acquisition of remote sensing images in postearthquake disaster areas and facilitates macroscopic analysis. In recent years, scholars both domestically and internationally have proposed a series of semiautomatic rapid landslide identification methods based on optical remote sensing images. Hu Wenmin et al. conducted landslide information extraction in the Lei Jia Shan landslide disaster area of Panping Village, Nanzhen Town, Changde City, Hunan Province, using the support vector machine (SVM) method based on the data from the Gaofen-2 (GF-2) optical remote sensing satellite [3]. In their study, Fu Junlin et al. used Landsat TM/OLI optical remote sensing satellite images to calculate the vegetation coverage values before and after the Qianjiangping landslide in Zigui County, Hubei Province. They analyzed the data from nine different time periods and found that the area of high vegetation coverage significantly decreased after a landslide, indicating a corresponding decrease in vegetation coverage after a landslide occurred [4]. Li Chenhui et al. employed high-resolution fusion images from the Gaofen-1 satellite and implemented a multiscale, object-oriented technique based on a multicondition threshold classification to rapidly interpret landslide information in the study area. This approach enabled the construction of hierarchical recognition rules for landslide identification [5]. Shao F et al. considered the introduction of the object-oriented conditional random field (CRF) optical remote sensing satellite image classification method to achieve the identification and extraction of landslide information in the study area [6]. Using Landsat series optical remote sensing satellite data, Wen Guangchao et al. proposed a method for the rapid identification of landslides by studying the spectral curve characteristics of landslides and other features in the target area, effectively extracting complete landslide information [7]. With the gradual improvement in the resolution of optical remote sensing image data, it has also found broader application in the dynamic monitoring of landslides. However, due to the harsh weather conditions in postearthquake areas, optical images are often hindered by cloudy weather and have low timeliness.

On the one hand, SAR (synthetic aperture radar) satellite technology is not affected by unfavorable factors such as rain, clouds and fog, and can image all day and in all weather, breaking through the limitations of optical images and gradually becoming the mainstream remote sensing technology means for earthquake emergency response and assessment. The method of landslide identification based on SAR images has been explored by many scholars: Nava L et al. and Mondini A C et al. interpreted landslides in SAR images based on landslide body geomorphic features [8,9]. Zheng Z et al. utilized interferometric synthetic aperture radar (InSAR) technology to determine the surface deformation of the study area and combined it with optical remote sensing imagery for manual interpretation of landslide information. Field verification confirmed the high accuracy of the InSAR monitoring results [10]. Yin W et al. proposed an adaptive identification method for potential landslide hazards based on multisource data, which was further improved and enhanced by incorporating InSAR technology to comprehensively identify potential landslides [11]. With the development of SAR sensors, SAR has gradually evolved from its original single mode of operation to a multiband, multipolarization, multiangle mode of operation, providing flexibility for seismic landslide identification applications [12]. On

the other hand, the normalized vegetation index (NVI) has been developed to provide a more flexible way to identify landslides. On the other hand, the normalized vegetation index has a high correlation with the leaf area index and biomass, which can reflect a luxuriant degree of surface vegetation and to a certain extent can indicate a change in surface vegetation cover [13]. The vegetation in Jiuzhaigou is well-developed and there are noticeable differences in vegetation cover between the pre- and postearthquake areas, particularly in regions with robust vegetation cover, which exhibit prominent variations.

Therefore, there are challenges in using optical remote sensing images due to the limitations imposed by natural cloud cover and the C-band wavelength of Sentinel-1 data, which makes it difficult to monitor and identify landslide hazards in densely vegetated mountainous areas. In terms of feature application, there is a lack of research on exploring vegetation activity changes, which prevents the full utilization of the rich information provided by high-resolution images. As a result, the accuracy of landslide information interpretation is not satisfactory and fails to meet practical application needs. In this paper, Jiuzhaigou is selected as the study area. Sentinel-1 satellite radar data are used to monitor postearthquake surface deformations in the research area. Combined with Sentinel-2 optical remote sensing image data, surface deformation information is introduced as a feature in the classification system. Furthermore, the changes in vegetation indices before and after the earthquake in the research area are analyzed in depth. A method for landslide information interpretation based on vegetation index difference using multisource data is proposed. This method aims to interpret seismic landslide disaster information. On the one hand, it provides new research ideas for landslide information interpretation. On the other hand, the results of landslide interpretation based on this method can assist in decision-making for railway route selection to a certain extent, allowing for proactive prevention and improving the efficiency and reliability of route selection, which holds great significance.

## 2. Study Area and Experimental Data

### 2.1. Study Area

Jiuzhaigou is located in the mountainous region of Southwest China, also known as Nanping County, which is part of the Aba Tibetan and Qiang Autonomous Prefecture in the northern part of Sichuan Province. With a total area of about 5300 square kilometers, Jiuzhaigou County has an overall step change in terrain from high in the northwest to low in the southeast, with an altitude drop of 3000 m and a humid plateau climate. In addition, the region is prone to geological disasters such as landslides and mudslides due to the constant development and frequent activity of fracture structures. Within 200 km of the epicenter, 142 earthquakes of magnitude 3 or greater have occurred in the last five years, with this one being the largest and most devastating. Jiuzhaigou has a humid plateau climate, with snow on top of the mountains all year round. It is divided by altitude into warm-temperate semi-arid, mid-temperate and cold-temperate monsoon climates; the average annual temperature is 12.7 °C, the average annual precipitation is 550 mm, the average annual sunshine is 1600 h and the average annual relative humidity is 65%. The high mountain areas above 4000 m in altitude are covered with snow all year round, with no vegetation cover, and the bedrock of the ridge is exposed. The external forces of the landscape mainly come from the weathering and erosion of glaciers and alpine freezing [14]. At lower elevations, mud flats and unstable slopes formed by landslides and avalanches can be observed. These areas are often active, with slow sliding occurring along the slopes over the years, posing a potential threat to the region. Below 3800 m in altitude, the landscape is characterized by dense vegetation, gullies and a network of rivers. The abundant vegetation primarily comprises forested slopes with high vegetation cover. During fall, the vertical variation in natural vegetation is most prominent, resulting in a picturesque scenery of overlapping greenery and golden layers of forest throughout the entire Jiuzhaigou scenic area. Additionally, there are exposed slopes that remain devoid

of vegetation due to natural factors such as faults and topography, including cliffs and landslides caused by previous landslides or eroding hillsides.

Jiuzhaigou is situated in the transitional region between the Qinghai–Tibet Plateau and the Sichuan Basin. It features a complex geological background characterized by the widespread presence of carbonate rocks, extensive folding fractures, active neotectonic movements and significant crustal uplift. The landscape is marked by the intersection of multiple geological media, resulting in diverse landforms and the formation of large-scale karst features, particularly calcium deposits. The region is located on the eastern edge of the Qinghai–Tibet Plateau and encompasses deep valleys and intricate topographic changes. It lies in the transition zone between the first and second steps of China's three major terrain steps, making it a frequent occurrence and regeneration area for various geological hazards, including landslides, collapses and mudslides. The geological background of Jiuzhaigou is highly intricate [15]. The boundary terrain of Jiuzhaigou is depicted in Figure 1, with the main valley displaying a "Y" shape. Numerous lakes, waterfalls and streams with calcareous beaches can be found within the valley.

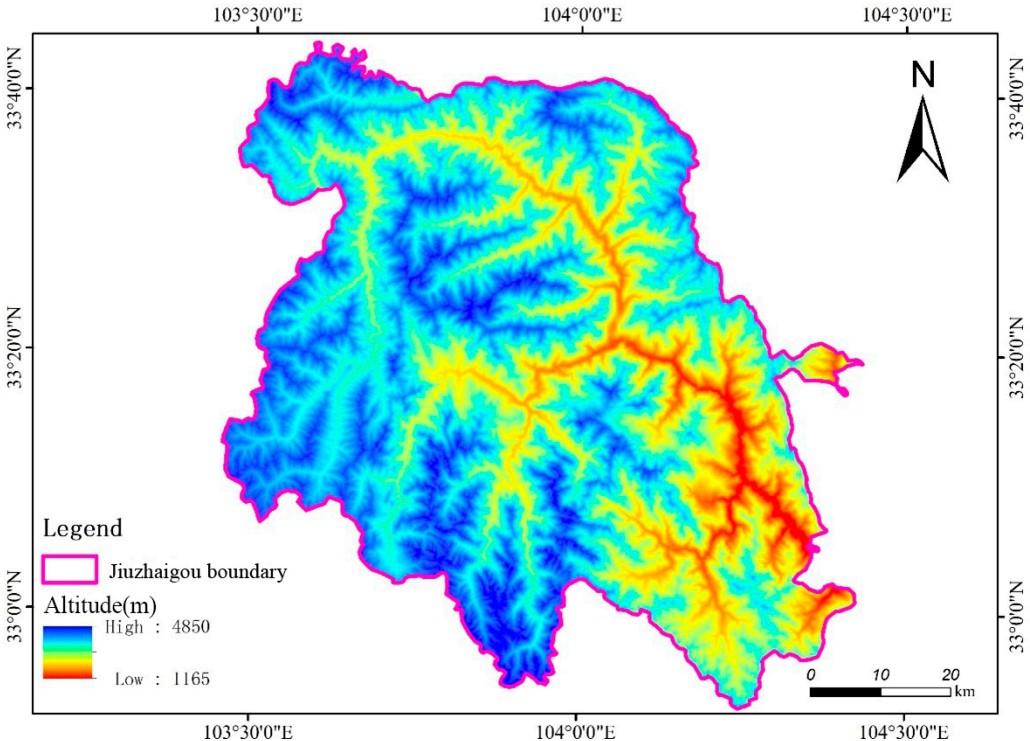

**Figure 1.** Image boundary topography of the study area.

## *2.2. Data Acquisition*

### 2.2.1. Optical Remote Sensing Data

The Sentinel-2 satellite system consists of high-resolution multispectral imaging satellites that revisit the same area every 5 days. It consists of two identical satellites, which are synchronized with each other in a sun-synchronous orbit at a phase separation of 180 degrees and an average altitude of 786 km. The position of each Sentinel-2 satellite in its orbit is determined using a dual-frequency global navigation satellite system (GNSS) receiver [16]. The satellites are equipped with a multispectral detection instrument (MSI) that captures images in 13 spectral bands, with ground resolutions of 10 m, 20 m and 60 m. These bands provide rich data that can be applied to various practical applications. Among optical remote sensing satellite data, Sentinel-2 data are unique in that they include three bands in the range of 680 to 760 nm, which is particularly useful for monitoring vegetation growth and health. The data acquired from Sentinel-2 are in Level-1C (L1C) format, which means they have not been corrected for atmospheric radiation and various

noises. In order to classify the data in remote sensing processing software and eliminate errors, further processing is required to obtain Level-2A (L2A) products, which mainly contain bottom-of-atmosphere corrected reflectance data. In this paper, the Level-1C product is radiometrically calibrated and atmospherically corrected using the sen2cor plug-in to obtain the Level-2A product. The specific parameters of the satellite are presented in Table 1. Sentinel-2 is primarily used for monitoring changes in the land environment, including vegetation growth and the extent of soil coverage. It can be applied to large-scale agricultural monitoring as well as the monitoring of natural disasters such as landslides.

**Table 1.** Sentinel II satellite band profile.

| Number of Bands | Wave Name | Sentinel-2A | | Sentinel-2B | | Resolution (m) |
|---|---|---|---|---|---|---|
| | | Central Wavelength (nm) | Bandwidth (nm) | Central Wavelength (nm) | Bandwidth (nm) | |
| 1 | Coastal aerosol | 443.9 | 20 | 442.3 | 20 | 60 |
| 2 | Blue | 496.6 | 65 | 492.1 | 65 | 10 |
| 3 | Green | 560 | 35 | 559 | 35 | 10 |
| 4 | Red | 664.5 | 30 | 665 | 30 | 10 |
| 5 | Vegetation Red Edge | 703.9 | 15 | 703.8 | 15 | 20 |
| 6 | Vegetation Red Edge | 740.2 | 15 | 739.1 | 15 | 20 |
| 7 | Vegetation Red Edge | 782.5 | 20 | 779.7 | 20 | 20 |
| 8 | NIR | 835.1 | 115 | 833 | 115 | 10 |
| 9 | Water vapour | 945 | 20 | 943.2 | 20 | 60 |
| 10 | SWIR–Cirrus | 1374.1 | 30 | 1377.1 | 30 | 60 |
| 11 | SWIR | 1620.1 | 90 | 1609.9 | 90 | 20 |
| 12 | SWIR | 2201.9 | 180 | 2185.7 | 180 | 20 |

Two main types of optical remote sensing image data were used in this study: Sentinel-2 (Sentinel II) data and Landsat 8 satellite data. The Sentinel-2 data were acquired from the European Space Agency's (ESA) Sentinel-2A satellite. This satellite has a total of 13 spectral bands, providing high-resolution imagery with an overall resolution of 10 m. The satellite revisits the same area every 10 days and the individual sensors sweep stripes that are 290 km wide. The specific band information for the Sentinel-2 data used in this study is presented in Table 2. It includes a single band with a resolution of 15 m, covering the wavelength range from 0.5 μm to approximately 0.75 μm. Additionally, the Sentinel-2 imagery has an overall resolution of 30 m and the strip width swept by individual sensors is 185 km.

**Table 2.** Optical remote sensing image data information.

| Data Type | Data Name | Resolution | Number of Bands | Date of Imaging |
|---|---|---|---|---|
| Landsat8 | LC81300372017192LGN00 | 15 m | 11 | 11 July 2017 |
| Sentinel-2A | L1C_T48SUB_A010969_20170729T035636 | 10 m | 13 | 29 July 2017 |
| Sentinel-2A | L1C_T48SUB_A011541_20170907T035630 | 10 m | 13 | 7 September 2017 |

### 2.2.2. Radar Image Data

The radar image data for the selected area were taken from the Sentinel I satellite, the first radar SAR data satellite launched by ESA, which is widely used for surface deformation monitoring, water system safety detection and mine subsidence [17]. The data consist of two polar-orbiting satellites, A and B, carrying synthetic aperture radar (SAR) sensors that are C-band and are active microwave remote sensing satellites. It consists of four main modes of operation: strip mode, interferometric broadband mode, ultrabroadband

mode and wave mode [18]. The Sentinel-1 satellite has two main advantageous features: (1) ultrahigh radiometric resolution, which can effectively enhance the accuracy of radar image parameter inversion; (2) higher revisit frequency and coverage performance, with a revisit period of 12 days for the same area by one satellite.

The type of Sentinel I data used is phase information data in progressive scan mode that can be used for interferometric processing, as shown in Table 3, for the resolution of SLC-1 level products. In this paper, lift-track SAR data pairs were acquired for two different time periods before and after the earthquake and the key information is shown in Table 4.

**Table 3.** Sentinel I satellite LOS directional resolution parameters.

| Mode | Resolution rg × az | Pixel Spacing rg × az | Number of Looks | ENL |
|------|--------------------|-----------------------|-----------------|-----|
| SM | 1.7 × 4.3 m to 3.6 × 4.9 m | 1.5 × 3.6 m to 3.1 × 4.1 m | 1 × 1 | 1 |
| IW | 2.7 × 22 m to 3.5 × 22 m | 2.3 × 14.1 m | 1 × 1 | 1 |
| EW | 7.9 × 43 m and 15 × 43 m | 5.9 × 19.9 m | 1 × 1 | 1 |
| WV | 2.0 × 4.8 m and 3.1 × 4.8 m | 1.7 × 4.1 m and 2.7 × 4.1 m | 1 × 1 | 1 |

**Table 4.** Radar satellite data parameters.

| Orbital Direction | Imaging Time (Main) | Imaging Time (Auxiliary) | Polarization Method | Spatial Baseline | Time Baseline |
|-------------------|---------------------|--------------------------|---------------------|------------------|---------------|
| Lift rail | 30 July 2017 | 23 August 2017 | VV | 72.031 m | 24 days |
| Lowering the track | 6 August 2017 | 18 August 2017 | VV | 64.085 m | 12 days |

### 2.2.3. Digital Elevation Model Data

A digital elevation model, or DEM for short, digitally simulates the surface terrain by using limited terrain elevation data to digitize the surface form of the terrain [19]. A DEM is a digital terrain model (DTM) that digitally describes the surface elevation, slope, aspect and rate of change of a slope and expresses the linear or nonlinear spatial combination of these factors.

ALOSDEM data were acquired for the study area, with ALOS data at 12.5 m surface resolution. The data are shown in Figure 2. Through operations such as projection transformation, a projection coordinate system was selected that is identical to the one that comes with the Sentinel II optical remote sensing image data itself: WGS_1984_UTM_Zone_48N planar right-angle coordinate system. Finally, the slope and slope direction raster maps of the study area were extracted, as shown in Figure 2. The maximum slope in the Jiuzhaigou area reached 83.1868°.

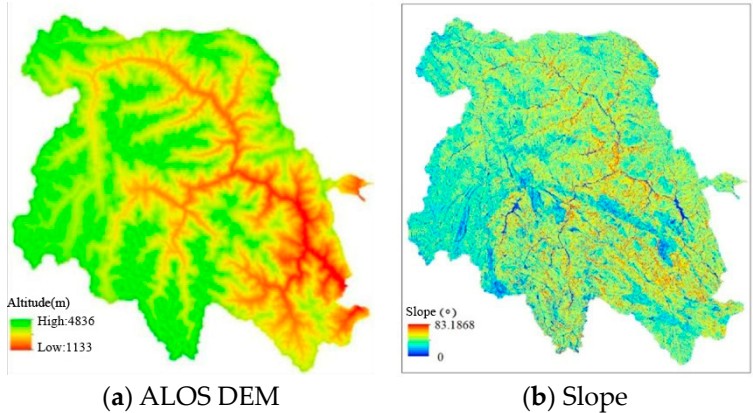

(**a**) ALOS DEM  (**b**) Slope

**Figure 2.** DEM and slope data.

### 2.3. Data Processing

During the operation of optical remote sensing satellites, optical remote sensing image data will inevitably be imaged by atmospheric radiation, spatial systems, time conversions

and feature spectra, thus reducing the image quality to a certain extent and generating errors. Therefore, preprocessing is required to eliminate errors as far as possible. In this paper, the preprocessing of optical remote sensing image data mainly includes radiation calibration and atmospheric correction, geometric correction, noise removal and image mosaic.

Multispectral images have a high number of image bands and contain a wealth of spectral information. As the spectral data are closely related to each other, the correlation of image bands may lead to complex data redundancy to a certain extent. Therefore, in the processing of remote sensing images, redundant data can slow down the data processing and thus reduce the efficiency, especially in sample selection, resulting in a redundant number of useless samples. Studies have shown that when the number of image training samples is kept at a specific value, the image classification accuracy tends to increase and then decrease in relation to the image band data.

Therefore, it is not "better to have more than one", but to consider the needs of the actual remote sensing image processing. For example, in this paper, we mainly used the vegetation information of the study area, so we chose the near-infrared band (B8), red band (B4), green band (B3) and blue band (B2) for image synthesis, which not only reduced the correlation interference of redundant data but also allowed the researchers to extract the feature information under the most ideal conditions. When selecting the bands, the following two main principles were considered: the information-rich band combination was chosen under the premise of correlation; the band combination was easier to identify the features. In summary, the statistical analysis of the image band data characteristics, combined with the existing research results, shows that the standard deviation of the image data bands directly determines the richness of the band information and there is a positive correlation between the two trends [20]. The standard deviation of the image bands directly determines the richness of the band information and there is a positive correlation between the two. Therefore, the larger the standard deviation of the bands, the richer the information of the bands and the better the application value in landslide information classification.

The B4 red band, B8 near-infrared band, B3 green band and B2 blue band of the Sentinel II remote sensing image were selected for image composition, image mosaic cropping of the study area and a 2% linear stretching of the image to enhance the information of the features on the remote sensing image. The preprocessed optical remote sensing image data are shown in Figure 3. The main feature information within the images acquired in this paper includes residential areas, vegetation, roads, snow-capped mountains, rivers, exposed rocks and landslides. The landslide feature information appears after the earthquake and is accompanied by a partial reduction in vegetation. Therefore, the remote sensing images of the study area showed significant changes before and after the earthquake.

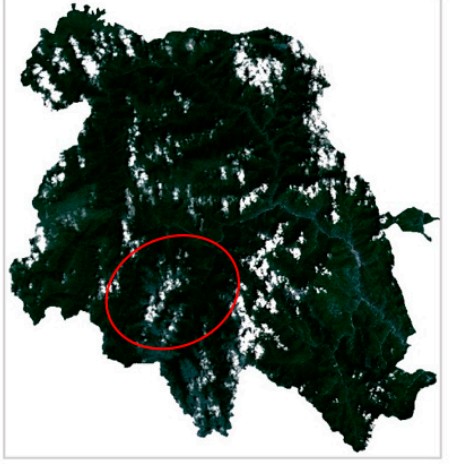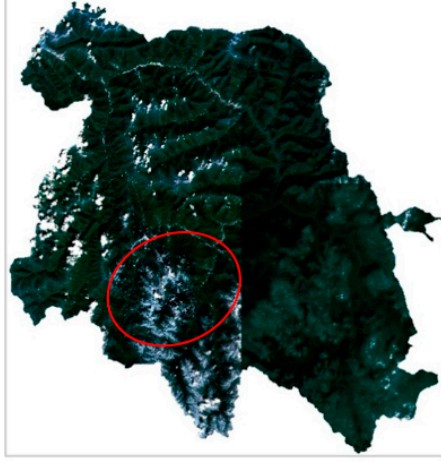

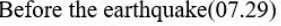
Before the earthquake(07.29)                    After the earthquake(09.07)

**Figure 3.** Optical remote sensing images.

## 3. Results

### 3.1. Postearthquake Landslide Deformation Information Extraction

The Sentinel-1A satellite carries a right-viewing radar for deformation monitoring in the Jiuzhaigou area using the two-orbit method, and the satellite's orbit and line-of-sight ground-range orientation are shown schematically in Figure 4.

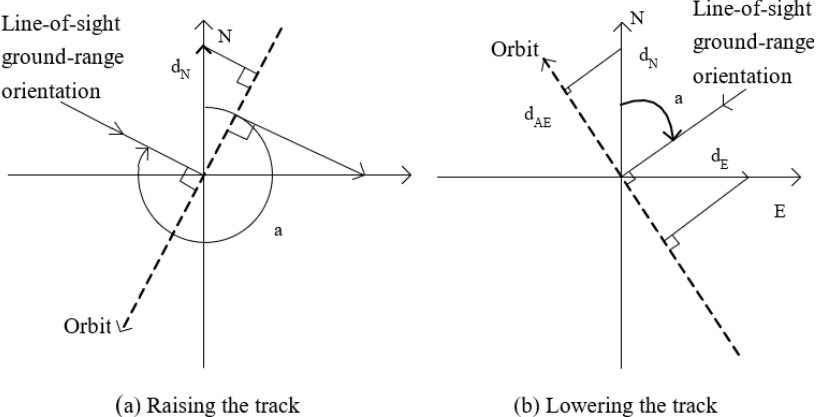

    (a) Raising the track          (b) Lowering the track

**Figure 4.** Satellite orbital orientation.

The Sentinel I radar image data before and after the earthquake were collected and the surface deformation information of the study area was extracted using the two-track differential interferometry technique. Firstly, the topographic phase difference images of the radar images were removed from the DEM data; secondly, the Goldstein adaptive filtering algorithm was used to eliminate the interference of atmospheric noise, body scattering and thermal noise in consideration of the dense vegetation and obvious topographic relief in the Jiuzhaigou area so as to effectively reduce the stacking mask during the imaging process; finally, the surface deformation information of the Jiuzhaigou area in the ascending and descending directions was obtained through phase decoupling and geocoding. The area of deformation information obtained was approximately 2655 km$^2$ and the results of the surface displacement data (in m) along the satellite line of sight are shown in Figure 5. From the figure, it can be seen that the maximum value of deformation is 0.0983336 m in the ascending orbit direction and the minimum deformation value is −0.277434 m. In the direction of lowering the satellite, the maximum value of deformation is 0.149176 m and the minimum value of deformation is −0.149176 m.

In order to better express the deformation information and visualize the research analysis, the classification result map was therefore reclassified in ArcGIS software and the classification result map is shown in 6. The results of the surface deformation information derived from the radar data in the ascending and descending orbital directions show some variability. In the ascending direction of the satellite, the surface deformation along the satellite line of sight in the study area mainly shows subsidence up to 28 cm, while at the same time, there is some uplift in the study area, with a maximum uplift of 9.8 cm. In the descending direction of the satellite, the surface deformation along the line of sight of the satellite in the study area mainly shows uplift, with a maximum of 14.9 cm; at the same time, there is a certain settlement in the study area and the maximum uplift is 11 cm.

Based on the research results, under the influence of uncertain natural factors, the unstable slope will be slowly deformed, especially in the geotechnical composite slope, the gap between the rock body becomes larger, the stress of the slope will increase, the original equilibrium and stability state will be destroyed and then the landslide disaster with a large area will be derived. Therefore, the large surface deformation information obtained by the D-InSAR technique can reflect the safety of the geological structure of the target area and the degree of damage under the influence of natural disasters to a certain extent and there is a positive correlation between the surface deformation variables and the state of existence of the mountain structure.

As the classification results of the lifting track deformation data differ to some extent, the study analyzed the existing literature to describe the variability of different factors [21]. The results were calculated by weighting the raster data and the weighted deformation results (m) are shown in Figure 5. The image reclassification was carried out on this basis and the results are shown in Figure 6. From the figure, it is concluded that the postearthquake deformation in the Jiuzhaigou area has a maximum settlement of 14.93 cm and a maximum uplift of 6.0 cm.

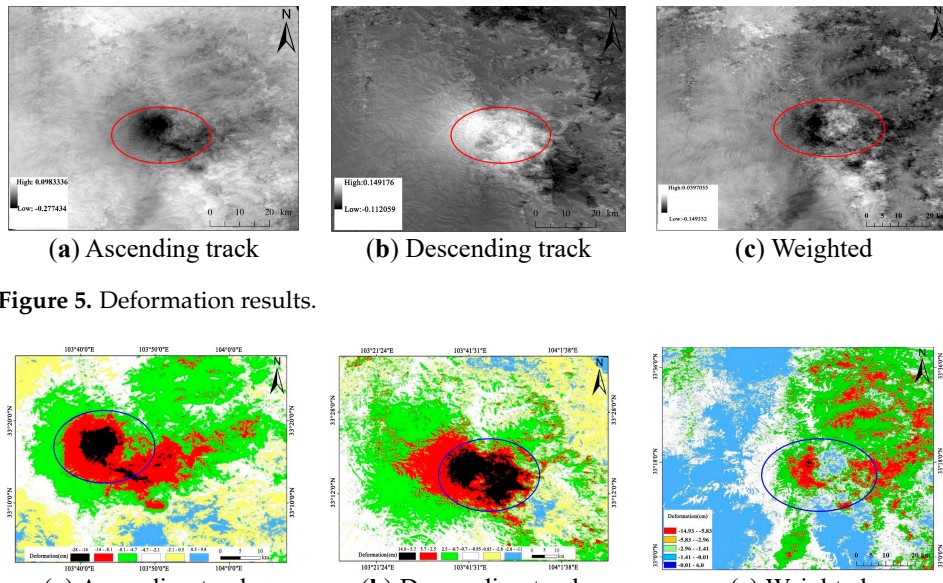

(**a**) Ascending track    (**b**) Descending track    (**c**) Weighted

**Figure 5.** Deformation results.

(**a**) Ascending track    (**b**) Descending track    (**c**) Weighted

**Figure 6.** Reclassification results.

### 3.2. Image Segmentation

Image segmentation refers to the segmentation of optical remote sensing images into different regions separated from each other according to the actual feature targets, which facilitates the grouping of features with the same attributes together and also provides a carrying platform for the subsequent refinement of image segmentation [22]. Karydas has developed an optimal scale selection method for image segmentation, which has been validated to show that it is successful and ensures the accuracy of image segmentation [23]. To summarize the current research status of image segmentation, the segmentation techniques are mainly divided into two types [24]: One is the a priori knowledge method, which is a top-down segmentation technique using a priori knowledge. This method is based on the premise of clarifying the properties of the target feature and finding a more consistent and reasonable segmentation model to extract the feature. The second is the data model method, which is a bottom-up segmentation technique using the acquired image data. This method is based on the spectral and spatial information of each feature of the optical remote sensing image and the statistics of each parameter to segment the image. This method is based on the premise that the target features of the image are not known and that there is no need to set a predefined classification target, which is also known as "generalised segmentation".

This method is based on a specific segmentation scale, which converts the target image elements of a feature on a high-resolution image into a small impact object that is "homogeneous to the maximum and heterogeneous to the minimum" while ensuring that the amount of information lost during image processing is small. The segmentation of the target feature varies between different segmentation scales. The advantage of multiscale segmentation is that the image elements of a feature target with different temporal and spatial attributes can find their "own home" using a specific scale. In multiscale segmentation, the homogeneity factor of the same image is determined by the spectral and shape

characteristics of the feature on the image, which depend on the smoothness and tightness of the image. The homogeneity factor has a weight of 1 and follows the principle of "local optimisation" in the segmentation process.

Multiscale segmentation generates attribute features of various feature objects that all characterize various feature targets at the same level. The smaller the segmentation scale, the more detailed and pure the generated object attributes are. However, when the segmentation scale area is infinitely small, the definition between objects of different feature classes becomes increasingly blurred. At the same time, when the segmentation scale becomes smaller, the segmentation objects will tend to be infinite, especially for the classification of remote sensing images of large areas, the segmentation results will be fragmented, which will seriously increase the computational workload of the computing equipment and reduce the extraction accuracy. Therefore, the segmentation scale should be "small but moderate".

At different scales, image information can be extended by image segmentation, thus representing image information at multiple scales. Each mutually distinct feature has different spectral characteristics of the feature, which correspond to their respective optimal segmentation scales. Relatively speaking, the optimal segmentation scale for an image is found without departing from the constraints based on a particular requirement. In practice, the optimal segmentation scale for a specific feature may or may not be applicable to other features, so a range of optimal scales needs to be determined for ease of use. In this paper, the optimal segmentation scale is determined using the local variance of image elements method, which is based on the eCognition software platform and is implemented using a scale estimation tool. The main principle of this method is to determine the optimal scale by calculating the local variance of each object on the image feature and the nonlinear variation in the rate of change curve of the heterogeneity of the image elements for each different feature. The optimal scale corresponds to the segmentation value when the mean variance of each object peaks and the rate of change in heterogeneity starts to show a decreasing trend. The implementation is shown in Equation (1).

$$ROC = \left[ \frac{L - (L-1)}{L-1} \right] * 100 \tag{1}$$

where $L$ is the mean local variance value corresponding to the current segmentation scale layer and $L-1$ is the mean local variance value corresponding to the next segmentation scale layer. The mean local variance and the homogeneity within the feature object show a negative correlation. The segmentation scales are determined as shown in Figure 7.

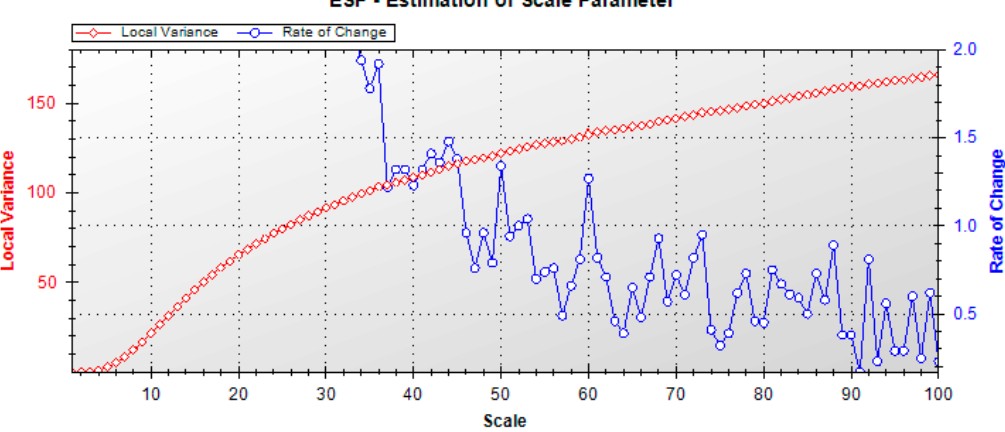

**Figure 7.** ROC/LV variation curve.

The Sentinel II optical remote sensing image was segmented using the local variance method and the final optimal segmentation scale range for this study area was obtained [50, 70]. Among them, the weight value of spectral features is 0.6, the weight value of shape

features is 0.4, the weight value of tightness features of the image is 0.2 and the weight value of smoothness features of the image is 0.2.

### 3.3. Vegetation Index Difference Analysis

The vegetation in the remote sensing images shows that the plant leaves show strong absorption in the red light band and strong reflectivity in the infrared light band. Based on this rule, the *NDVI* (normalized difference vegetation index) extracts vegetation information by combining the linear operations of the red band ($B_3$) and the infrared band ($B_4$) of the image.

$$NDVI = \frac{B_4 - B_3}{B_4 + B_3} \qquad (2)$$

The vegetation index difference combines the vegetation cover change characteristics of the Jiuzhaigou area before and after the earthquake and the vegetation index characteristics of the same image element at different times are differenced to obtain the vegetation cover change information in the postearthquake period. In this paper, combining the characteristics of good vegetation cover in the Jiuzhaigou area, the optical remote sensing image information of Sentinel II was selected for two time periods before and after the earthquake. The vegetation index information of the different periods was calculated and visualized using Equation (2) and the results are shown in Figure 8.

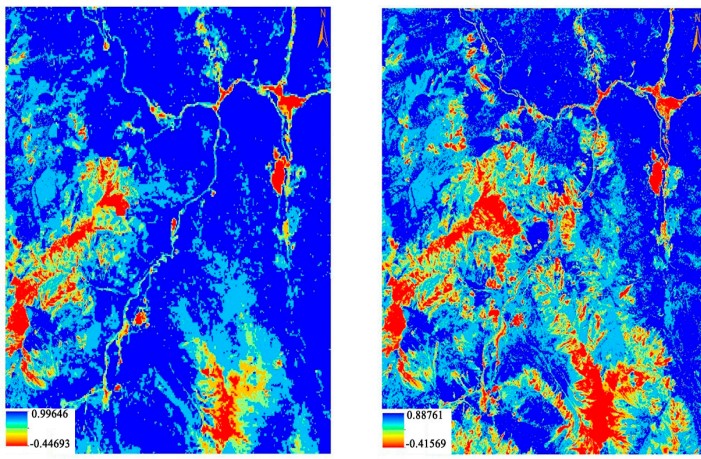

Before the earthquake (07.29)     After the earthquake (09.07)

**Figure 8.** Vegetation index (partial).

The main features extracted from the acquired images include residential areas, vegetation, roads, snow-capped mountains, rivers, exposed rocks, landslides and so on. The minimum threshold for vegetation index changes is −1, while the maximum threshold is 1. Therefore, when the vegetation index tends towards −1, it indicates that the objects in that area exhibit weak absorption and strong reflection of visible light. The types of objects in such areas may include water bodies, clouds or snow. When the vegetation index is greater than 0 and tends towards 1, it indicates the presence of vegetation cover and there is a positive correlation between the vegetation index and vegetation coverage. When the vegetation index is 0, it suggests that the objects in that area are likely to be exposed rocks or bare land. As shown in Figure 8, the areas displayed in deep blue represent locations with good vegetation cover, indicating areas with vegetation coverage.

As the earthquake occurred during the last months of vegetation development, June to September, there was a significant difference in the characteristics of the lush vegetation in the region before and after the earthquake, which was also expressed in the optical remote sensing images. Therefore, after comprehensive consideration of the high vegetation cover characteristics of the Jiuzhaigou region, the vegetation index difference data feature was

introduced to reflect the pre- and postearthquake vegetation changes in the region, thus using this distinctive feature to achieve landslide information extraction.

The normalized difference vegetation index (*NDVI*) is an important activity indicator reflecting the health and growth status of vegetation and is the most widely used indicator in vegetation remote sensing applications. After considering the high vegetation cover characteristics of the Jiuzhaigou area, the vegetation index difference data feature was introduced to reflect the pre- and postearthquake vegetation changes in the area, so that this obvious feature could be used to extract landslide information. The vegetation index features are shown in Figure 9.

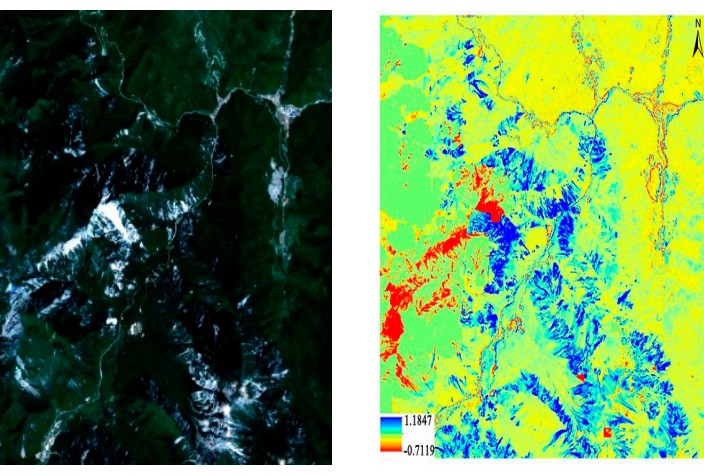

Postearthquake (09.07)          Difference in vegetation index

**Figure 9.** Local comparison of remote sensing imagery and vegetation index differences.

From Figure 9, it can be observed that the vegetation index difference image primarily consists of deep blue, green, yellow and red colors. The deep blue areas indicate a vegetation index difference greater than 0, mainly reflecting the landslide information that occurred in the area after the earthquake. The green and yellow areas primarily reflect the pre-existing features in the study area. The red areas indicate a vegetation index difference of less than 0, representing the changes in the location of the pre-existing features in the study area caused by the earthquake. These changes indicate a relative decrease in the target objects.

*3.4. Landslide Information Interpretation*

3.4.1. Classification of Affiliation

Most of the remote sensing images after multiscale segmentation are mixed images, which cannot be discriminatively assigned to a feature class. In this paper, fuzzy classification is chosen to continue the classification of multiscale segmented images. Fuzzy classification is performed by analyzing the affiliation relationship between the segmented object features and the spectral and spatial features of the feature to be classified and assigning a fuzzy value within the range. Fuzzy classification is also known as affiliation classification [25]. Thus, fuzzy classification is the process of determining the affiliation between the object to be segmented and the target feature to be extracted to determine the object to be segmented and the need for the study by using a specific affiliation function to calculate the affiliation value and then using the calculated value to fuzzy the feature value and determine the uncertain spatial relationship between the features and the image elements. The essence of the fuzzy rule is an "if-then" conditional execution statement. For a particular segmented feature object, a segmentation threshold is determined and if the affiliation is greater than this threshold, the object belongs to the class with the highest affiliation; if it is less than this threshold, the object is excluded from the target feature. The affiliation function mainly has a triangular affiliation function, trapezoidal affiliation function and affiliation function. Among them, the affiliation function means that the affiliation

of all objects within the distribution range is 1, discarding objects that are not within the range, thus greatly simplifying the number of fuzzy rules. The Gaussian affiliation function performs well in terms of functional symmetry and smoothness and if the affiliation degree is positive, the affiliation function can be approximated infinitely.

Considering the relevance and spatial and temporal characteristics of the features, this paper selects an affiliation function classification function based on the spectral characteristics of the features, establishes recognition rules and classifies them. As the characteristics of the seismic landslide studied in this paper are very different from the shapes and spectral features of other surrounding features, combined with the affiliation function classifier and drawing on previous classification studies, the affiliation function is selected to achieve the decoding of landslide information, which can effectively reduce the learning cost and streamline the number of fuzzy rules.

### 3.4.2. Selection of Object Attribute Features

Optical remote sensing images are segmented at multiple scales, resulting in several segmentation units that are different from traditional pixels and best to reflect the abstract feature information. By analyzing the relationship between the target feature and the segmented object, a variety of suitable feature characteristics are selected to characterize the feature. Therefore, it is necessary to select suitable features to distinguish landslide information from other information. Therefore, in image segmentation, feature recognition is actually the analysis and recognition of various features of representative features. In this paper, the reflection of the obtained surface deformation information on the seismic landslide information is comprehensively analyzed and the surface deformation information is considered as a feature value introduced into the establishment of recognition rules combined with the overview of the Jiuzhaigou area and experimental data. Based on the postearthquake Sentinel II image data, this paper introduces the slope and brightness feature factors as well as the vegetation index difference and the deformation information extracted from the study area using radar remote sensing images. In this paper, the constructed vegetation index difference data are introduced into the landslide information classification system, which is selected to combine the postearthquake vegetation index, slope and surface deformation of the study area, so as to establish the classification recognition rules. The threshold range for each feature was derived through the feature value interface of the econ software, as shown in Table 5.

**Table 5.** Landslide information extraction feature thresholds.

| Object | Feature | Threshold Range |
|---|---|---|
| | Vegetation index difference | $NDVI_x > 0.22$ |
| | Postearthquake images | $-1 < NDVI < 0.6$ |
| Landslide information | Slope (°) | >12 |
| | Surface deformation (m) | < >0 |
| | Brightness | <3000 |

### 3.5. Accuracy Evaluation Method

The confusion matrix method was applied to the accuracy evaluation method of remote sensing image classification results [26,27]. This method checks all the image elements in the classification sample area and counts the degree of confusion between the categories in their classification maps and the actual categories, which can objectively evaluate the degree to which the image elements in the classification results have been correctly categorized, and its definition formula is as follows:

$$M = \begin{bmatrix} m_{11} & m_{12} & \cdots & m_{1n} \\ m_{21} & m_{22} & \cdots & m_{2n} \\ \cdots & \cdots & \cdots & \cdots \\ m_{n1} & m_{32} & \cdots & m_{nn} \end{bmatrix} \tag{3}$$

In the confusion matrix, the elements on the diagonal are the number of correctly categorized samples and the elements on the off-diagonal are the number of misclassified samples. The evaluation metrics for the confusion matrix are the Kappa coefficient, which is a metric used to determine the agreement or accuracy between two images by multiplying the total number of all true reference pixels ($N$) by the sum of the diagonal of the confusion matrix ($X_{KK}$) and subtracting the product of the number of true reference pixels in a class and the total number of classified pixels in that class, then dividing it by the square of the total number of pixels subtracted by the square of the total number of pixels subtracted by the square of the total number of pixels subtracted by the square of the total number of pixels subtracted by the square of the total number of pixels in a class. The sum of the product of the total number of true reference image elements in a category and the total number of classified image elements in that category is summed over all categories. The Kappa coefficient is calculated as follows [28]:

$$k = \frac{N\sum_k x_{kk} - \sum_k X_{k\Sigma^x\Sigma k}}{N^2 - \sum_k X_{k\Sigma^x\Sigma k}} \tag{4}$$

The Kappa coefficient is defined as shown in Table 6.

**Table 6.** Defined range of Kappa factors.

| Kappa Factor | Precision |
|---|---|
| 0.0~0.20 | Very low consistency |
| 0.21~0.40 | General consistency |
| 0.41~0.60 | Medium consistency |
| 0.61~0.80 | Highly consistent |
| 0.81~1 | Almost identical |

## 4. Discussion

### 4.1. Analysis of Results

Firstly, lift-track radar data along the line-of-sight direction of the Sentinel-1 satellite before and after the earthquake in the Jiuzhaigou area were collected and the surface deformation information of the study area after the earthquake was obtained using the two-track method. The surface deformation results of the elevated and lowered tracks were analyzed together and different weighting values were assigned to the data through a weighting analysis to achieve a weighted superposition of the surface deformation information. The results show that the postearthquake deformation in the Jiuzhaigou area is mainly subsidence, with a value of up to 14.93 cm and some areas with about 6.0 cm of uplift. Under the influence of uncertain natural factors, unstable slopes exhibit slow deformation, especially in geotechnical composite slopes where the voids between the rocks become larger, increasing the stress on the slope and destroying the original equilibrium and stability, which in turn leads to large landslide hazards. Therefore, significant surface deformation information can reflect the safety of the geological structure of the target area and the degree of damage under the influence of natural disasters to a certain extent, and there is a positive correlation between surface deformation and the existence of the mountain structure. This paper considers the introduction of surface deformation information as a feature value to establish classification rules and, for the first time, combines radar data and optical remote sensing data to achieve landslide information interpretation under multiple sources of data. Secondly, the vegetation in the Jiuzhaigou area is well developed and covers a wide range of areas, especially the areas below 3800 m in elevation where the vegetation is luxuriant, the valleys are crisscrossed and the river network is dense. Therefore, the vegetation cover changes and growth status of the study area before and after the earthquake were combined to obtain a complete pre- and postearthquake vegetation index based on Sentinel II optical remote sensing images and the vegetation index difference values were calculated for the time interval to accurately reflect the changes in

vegetation activity. Therefore, the vegetation index difference data features are considered to be introduced into the classification system to achieve landslide information interpretation and "tailor-made" landslide geological hazard interpretation for the Jiuzhaigou area. Finally, the paper uses 60 as the best image segmentation scale, combined with the spectral factor, shape factor, compactness and smoothness with a weight of 0.5, to extract and study the postearthquake landslides in the Jiuzhaigou area using multiscale segmentation fuzzy classification based on vegetation index difference and decodes the landslide distribution information. More than 800 landslide records with an area of 12.034 km² were deciphered using ArcGIS software and manual visual interpretation. The results are consistent with the findings of the Sichuan Earthquake Bureau [1]. The landslide extraction results are shown in Figure 10. Based on more than 800 landslide records, the landslide distribution map of the Jiuzhaigou area was obtained using ArcGIS software. The map displays a total of 164 landslide points, as shown in Figure 11.

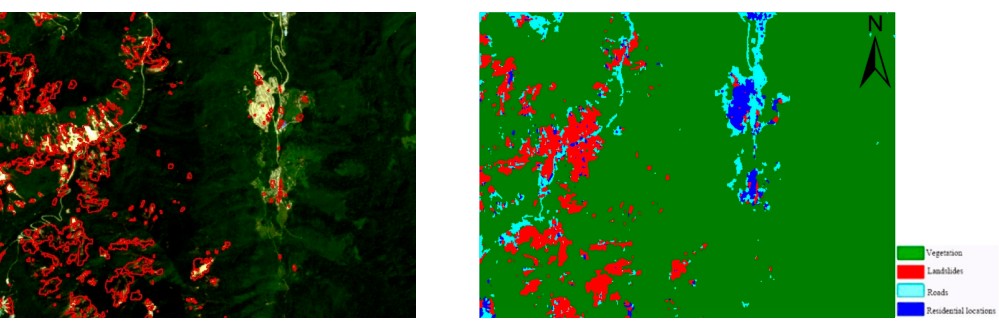

**Figure 10.** Local classification of landslide information extraction results.

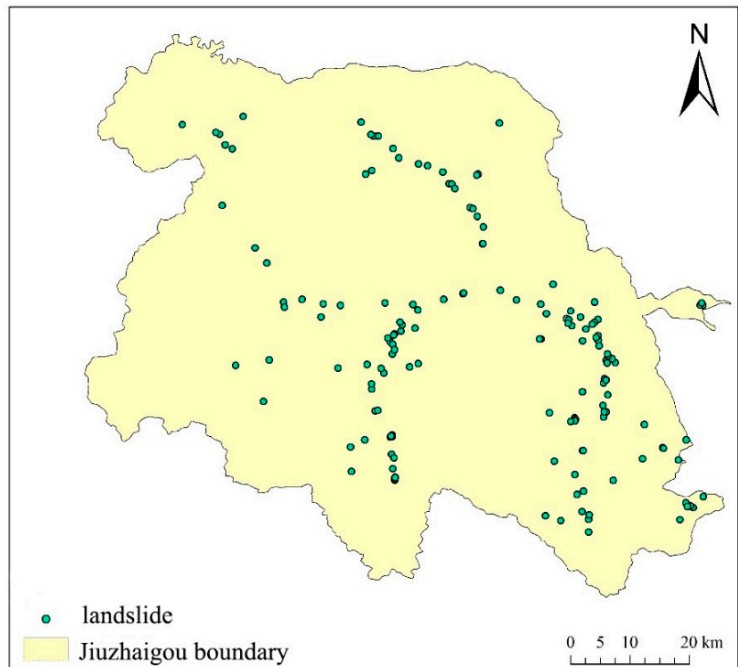

**Figure 11.** Landslide distribution point map.

*4.2. Accuracy Evaluation*

For the proposed classification method decoding results, 500 objects were randomly selected as test data for analysis using ArcGIS software and the accuracy evaluation was achieved by confusion matrix. The results are shown in Table 7, which shows that the overall accuracy of the classification is 92.6% and the Kappa coefficient is 0.8763. The Kappa coefficient usually falls between 0 and 1 and the Kappa coefficient is defined in

Table 6, which shows that the Kappa coefficient of the classification is 0.8763, which is between 0.81 and 1. The results of the experiments have excellent accuracy and are "Almost identical".

**Table 7.** Confusion matrix for classification results.

| Reference Data | Actual Classification Data | | | | Total Reference |
|---|---|---|---|---|---|
| | Vegetation | Landslides | Roads | Residential Locations | |
| Vegetation | 218 | 1 | 0 | 1 | 220 |
| Landslides | 12 | 89 | 7 | 7 | 115 |
| Roads | 0 | 3 | 71 | 3 | 77 |
| Residential locations | 0 | 7 | 2 | 79 | 88 |
| Total classification | 230 | 100 | 80 | 90 | 500 |
| Overall accuracy | 92.6% | | | | |
| Kappa factor | 0.8763 | | | | |

From the confusion matrix of classification results, it can be obtained that out of the 100 selected landslide points, 3 points were misclassified into road information, while 7 points were misclassified into residential points. This is due to the blurred boundaries of some landslides occurring near residential points as well as roads, which are manifested in the optical remote sensing images, i.e., landslides occurring near residential points and causing damage to roads and, thus, there are inevitable errors in the interpretation of the landslide information of the image. Based on the results of this experiment, the analysis concluded that the high-resolution remote sensing image interpretation based on vegetation index difference proposed in this paper has good performance, the classification accuracy reaches the expected standard and the classification effect is obvious. By using radar satellite earth observation technology to obtain information on the large-scale surface deformation of the study area, combined with information on the vegetation cover changes in the study area, we can achieve feature classification and identification of the study area as well as the postearthquake landslide information interpretation based on these two data sources from different sources and different periods. On the one hand, the method is rich in data sources and can ensure good experimental accuracy; on the other hand, the vegetation in the Jiuzhaigou area is well developed, especially before and after the earthquake, the vegetation cover in the area where the landslide occurred deteriorated and the difference in vegetation index can reflect the change in vegetation activity during that period; thus, the method can take advantage of this distinctive feature and make full use of optical remote sensing image information to achieve a good combination of pre- and postearthquake data sources. The complex topography of the Jiuzhaigou region, with its many layers, inevitably results in blurred boundaries of the target features on the optical remote sensing imagery, which can easily lead to misclassification or omission of feature information points. In order to further improve the classification accuracy, it is necessary to conduct more detailed studies on the spectral information, neighborhood, background, topography and geological conditions of the images in the study area in order to obtain more effective results. The sample selection method can also be improved so that more representative samples can be selected more scientifically and effectively.

## 5. Conclusions

Using the Jiuzhaigou area as the study area, a substantial exploratory study of landslide geological hazards was conducted for the first time by analyzing the *NDVI* difference before and after the earthquake and using radar imagery and optical remote sensing imagery. The following main conclusions were drawn.

(1)    The surface deformation information of the study area was extracted using the D-InSAR (differential synthetic aperture radar interferometry) technique along the ascending and descending orbits. Weighting analysis was performed to assign different weight values to the data, enabling weighted superposition of the surface deformation information. The final results indicate that the maximum subsidence after the earthquake in the Jiuzhaigou area was 14.93 cm, while the maximum uplift was 6.0 cm.

(2)    By introducing the vegetation index difference data and surface deformation information into the establishment of landslide identification rules and finally extracting the landslide information of the study area using fuzzy classification, a total of about 12.034 km$^2$ of the landslide area in the study area was obtained and 164 landslide point distribution maps were obtained and the overall accuracy of this classification was 92.6%, with a Kappa coefficient of 0.8763.

(3)    The article proposes a landslide information interpretation method based on the difference in the vegetation index under multisource data; the effective realization of this method provides a new research idea for landslide information interpretation on the one hand; on the other hand, the results of landslide interpretation based on the method can provide a reference for the selection of railroad lines in mountainous areas to a certain extent, such as for different scales of landslides, it can be taken to bypass or effective engineering measures to ensure that the line will not be affected by the landslides and ensure the safe operation of the railroad line.

The research methodology presented in this paper is most suitable for complex vegetated mountainous areas. However, it may be challenging to observe the impact of vegetation index difference as a characteristic parameter in areas with low vegetation coverage. The paper also lacks mutual validation between the measured data and InSAR results. Additionally, the SAR data required for InSAR technology could be replaced with ALOS PALSAR-2 data, which have stronger penetration capabilities, but these data are relatively scarce in archives.

**Author Contributions:** Writing—original draft preparation, J.Q.; funding acquisition, J.Q.; writing—review and editing, L.L. and J.L.; visualization, S.W.; supervision, Y.Z. All authors have read and agreed to the published version of the manuscript.

**Funding:** This research was supported by a grant from the Technology Research and Development Plan Key project of China National Railway Group Co., Ltd. (No. P2022G054). This support is much appreciated.

**Institutional Review Board Statement:** Not applicable.

**Informed Consent Statement:** Not applicable.

**Data Availability Statement:** Not applicable.

**Conflicts of Interest:** The authors declare no conflict of interest.

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
