# Peer review of "Improving Railway Alignment Selection in Mountainous Areas with Complex Vegetation: A Multisource Data Landslide Identification Approach for Assisted Decision-Making Research"

_sustainability, doi:10.3390/su151411388_

Round 1
Reviewer 1 Report
This paper proposes a new method of landslide interpretation in complex vegetated mountainous areas combined with multivariate remote sensing data, introducing normalized vegetation index difference and surface deformation information as the feature values for landslide information interpretation, which can be seen from the conclusion section of the article that the interpretation accuracy is high and aids in decision-making for railroad alignment options in complex vegetated mountainous areas. It has a good research idea. The following are some of my comments that emerged during the reading process.
1) The first and second conclusions in the conclusion section do not have a quantitative representation of the data obtained from the experiment and are suggested to be revised.
2) The third conclusion in the conclusion section should be as concise as possible.
3) The figure name in line 490 is wrong, suggest to change it.
4) There is a writing error in line 667, suggest to correct it and check for errors elsewhere.
Reviewer 2 Report
The manuscript is original and describes well the problematic of adequatley identify the distribution of landslides for future railway infraestructure.
The research design, questions, hypotheses and methods are clearly stated, and the results clearly presented. The figures are of quality and conclusions are suported by the results.
My only issue with this paper is that authors referenced to many studies from its country, and I strongly recommend to cite other studies around the world.
Reviewer 3 Report
I read the article titled " Aided decision making research on railroad alignment selection based on multi-source data landslide identification method for complex vegetation mountainous areas".
Firstly, the recommendation for the title of the manuscript: There are two different jobs in this comprehensive article. One is "Railroad Alignment Selection" and the other is "Landslide Identification Method". These tasks are carried out separately in the paper, but one task is to develop an innovative methodology for landslide identification. Suggestion title: “Improving Railway Alignment Selection in Mountainous Areas with Complex Vegetation: A Multi-Source Data Landslide Identification Approach for Assisted Decision Making Research.
Below are my header-title and line-numbered comments.
Introduction
The "Introduction" section is well written overall.
Line 40, Has a double point.
Line 41-50: Furthermore, in the introduction, would you like to say something about whether "fault lines" are indeed a landslide factor? The landslides that occurred after a major earthquake in the past months could be seen with the Landsat satellite image.
Link: https://earthobservatory.nasa.gov/images/151016/landslides-in-turkiye
Line 59-61: Read and refer to the most recent studies integrating machine learning into the identification of landslide areas. Please see:
Analysis of Conditioning Factors in Cuenca, Ecuador, for Landslide Susceptibility Maps Generation Employing Machine Learning Methods. Land 2023, 12, 1135. https://doi.org/10.3390/land12061135
Line 88, Please use [8-9] reference style.
The research question is well-defined.
The material-method section of the paper should provide information on how the fuzzy classification was integrated into the study. The information in Section 3.4 is a material-method statement rather than a conclusion.
Section 3.2. Material-method sentence. Why the ROC equation is mentioned in the results? Please make a great effort to organize the article in the appropriate sections. The reader has to agonize to understand.
Also, Can distance from fault lines be considered a factor in the formation of a landslide?
A sample study: https://link.springer.com/article/10.1007/s11629-022-7685-y
Of course, Fault may not be added as a factor. You are the one who knows best all the conditions in the current study area. However, if there is an omission, please refer to it as a "limitation for the case study" in the "discussion" section.
Why is Section 4.2 not in the material method?
You need to give more detailed information about Figure 11 in the material-method section. If a classification-based system is presented with a fuzzy approach, you should write this in the material method.
How are lithology and land cover datasets included in the modeling? Or included?
What are the limitations of your work on landslide classification with a classical fuzzy-based approach? Is your work flawless?
How do future or current approaches evolve in landslide risk or occurrence mapping?
What is the ability of machine learning algorithms to spatially represent a complex phenomenon such as landslides?
The conclusions are too long.
What is the message to take home?
Sincerely yours,
Reviewer 4 Report
The method used in this study is commendable because it uses the difference of normalized difference vegetation indices and surface deformation information as feature values through multi-source remote sensing data, which strengthens the reliability of the study classification results. The following are some of my comments that emerged during the reading process. They are suggested to be revised.
1) In line 251, the term "slope aspect" should be "slope aspect".
2) In line 259, how is the slope aspect value known, no clear evidence should be deleted.
3) There are some problems with English grammar, and the terminology in the text should still be carefully checked and revised.
4) Recent relevant references are missing and the authors should include some.
English is acceptable
Round 2
Reviewer 3 Report
I am not used to formulas (Kappa or ROC) in the "3. Results" section. This affects the reading of the article in general. Apart from that, the authors provided the necessary answers and rebuttals to the questions I asked.
Materials and Methods: They should be described with sufficient detail to allow others to replicate and build on published results. New methods and protocols should be described in detail while well-established methods can be briefly described and appropriately cited. Give the name and version of any software used and make clear whether computer code used is available. Include any pre-registration codes.
Results: Provide a concise and precise description of the experimental results, their interpretation as well as the experimental conclusions that can be drawn.
If the contents of the material-methods and results sections are in accordance with the journal policy and instructions (https://www.mdpi.com/journal/sustainability/instructions), I recommend that the article be accepted for publication under the editor's responsibility.
Best regards